# Airway Findings in Patients with Hunter Syndrome Treated with Intravenous Idursulfase

**DOI:** 10.3390/jcm12020480

**Published:** 2023-01-06

**Authors:** Richard De Vuyst, Elizabeth Jalazo, Tamy Moraes Tsujimoto, Feng-Chang Lin, Joseph Muenzer, Marianne S. Muhlebach

**Affiliations:** 1Department of Pediatrics, University of North Carolina, Chapel Hill, NC 27599, USA; 2Department of Biostatistics, UNC Gillings School of Global Public Health, Chapel Hill, NC 27599, USA; 3Marsico Lung Institute, University of North Carolina, Chapel Hill, NC 27599, USA

**Keywords:** tracheopathy, Hunter syndrome, lysosomal storage disease, bronchoscopy, airway severity score, enzyme replacement therapy

## Abstract

People with Hunter syndrome are known to be affected by a variety of airway pathologies. Treatment of Hunter syndrome with the enzyme replacement therapy (ERT) idursulfase is now the standard of care. However, it is not known how ERT changes the progression of airway involvement. To evaluate this, we performed a retrospective analysis of bronchoscopies performed on children with Hunter syndrome who were part of intrathecal ERT trials. Findings for airway pathology were extracted from bronchoscopy reports and analyses were performed for cross-sectional and longitudinal changes in airway disease. One-hundred and thirty bronchoscopies from 23 subjects were analyzed. Upper airway disease (adenoid hypertrophy and/or pharyngomalacia) was reported in 93% and 87% of bronchoscopies, respectively. Laryngeal abnormalities were recognized in 46% of cases. There were lower airway (tracheal and or bronchial) findings in 64% of all bronchoscopies and prevalence increased with age. Evaluations over time adjusted for repeat evaluations showed that increasing airway involvement was associated with older age (*p* = 0.0007) despite ongoing ERT. No association was discovered between age of intravenous ERT initiation and progression of airway disease. Individuals with Hunter syndrome who are receiving intravenous enzyme replacement therapy showed the progression of airways disease supporting the need for regular airway monitoring and intervention.

## 1. Introduction

Mucopolysaccharidosis II (Hunter syndrome, MPS II) is an X-linked recessive lysosomal storage disease with multiple organ involvement and variable progression. It is caused by a deficiency of the enzyme iduronate-2-sulfatase [1]. This results in the accumulation of the glycosaminoglycans (GAG) dermatan sulfate and heparan sulfate within lysosomes which results in gradual physiologic and anatomic dysfunction. GAG accumulation leads to progressive cardiac valvular disease, pulmonary dysfunction, and skeletal deformities. Two thirds of individuals with MPS II have the severe or neuronopathic form of the disease which manifests as early cognitive impairment with progressive neurological decline; without treatment, this leads to death in the first or second decade of life. Breathing impairment is recognized as a major cause of morbidity and mortality, with respiratory failure representing a common cause of death in affected patients [2,3,4]. Upper airway complications are a common presenting problem and individuals are frequently treated for upper airway problems prior to diagnosis of MPS II [5,6,7]. The upper airway manifestations include adenoidal and tonsillar hypertrophy, macroglossia, and laryngomalacia [6,7,8,9]. Lower airway dysfunction includes tracheomalacia, bronchomalacia, narrowing of airways due to deposition of GAG, and restrictive lung disease due to decreased chest wall compliance [6,8]. 

The development of the intravenous (IV) enzyme replacement therapy (ERT) idursulfase has provided a treatment option for individuals with MPS II since 2006. Intravenous idursulfase has been shown to be well tolerated and in clinical trials demonstrated decreased liver and spleen sizes and decreased urinary GAG levels, and either stabilized or improved several clinical parameters, including walking ability (6-min walk test), lung volume as measured by forced vital capacity (FVC), left ventricular mass index, and joint range of motion (shoulder) [4,10,11]. However there has been a paucity of research into intravenous idursulfase effects on specific airway findings associated with MPS II.

We had previously shown a high incidence of airway complications in individuals naïve to IV ERT [6]. We hypothesized that ERT in MPS II would prevent progression of airway pathology. The aims of this study were to describe the incidence and longitudinal course of airway pathology in a population of individuals with the severe phenotype of MPS II treated with IV ERT.

## 2. Materials and Methods

### Study Setting and Subjects

This study is a retrospective analysis of individuals with the severe or the neuronopathic form of MPS II, who participated in the phase I/II (NCT00920647 and NCT01506141) or phase II/III (NCT00937794 and NCT00920647) intrathecal enzyme replacement clinical trials at the University of North Carolina at Chapel Hill site since 2011. As part of their annual clinical trial evaluations requiring general anesthesia, the participants underwent airway evaluation by flexible bronchoscopy and if needed, fiberoptic intubation to decrease the risk of airway complications with general anesthesia. Recruitment criteria for the phase I/II MPS II intrathecal enzyme replacement clinical study included individuals’ ages 3 to 17 years, cognitive impairment and at least 6 months of IV idursulfase at enrollment into the study. Recruitment criteria for the phase II/III MPS II intrathecal enzyme replacement clinical study included individuals’ ages 2 to 17 years, cognitive impairment and at least 4 months of IV idursulfase at enrollment into the study. All subjects were treated with weekly IV and monthly intrathecal ERT throughout the period in which data for the current study were collected.

Clinical information collected from medical records included patient demographics, age at MPS II diagnosis, age at first bronchoscopy, age at IV ERT initiation, number of bronchoscopies and age at each bronchoscopy, history of prior airway surgery, and airway findings on bronchoscopy.

Ethics

The airway findings analyzed in this report were from MPS II patients who were enrolled in intrathecal ERT studies at the University of North Carolina at Chapel. These studies were approved by the University of North Carolina Institutional Review Board (IRB) with consent obtained from the parents or guardians of each subject. For this retrospective airway study, IRB approval at the University of North Carolina was obtained (20-2178) but was deemed exempt from consent due to posing no more than a minimal risk to the subjects. 

Assessment of Airway Disease

To analyze the relationship between ERT and the degree of airway involvement, we developed a grading system of airway severity for statistical analysis. The airway severity score system was developed to include the wide variety of airway pathology seen in MPS II individuals while preventing the overrepresentation of any single anatomic area. To determine how each area of the airway contributed to progression of anatomical changes, bronchoscopy findings were analyzed for the three anatomic divisions: upper, laryngeal, and lower airway. Findings were typically expressed as mild, moderate, or severe in bronchoscopy reports and we graded all findings on a scale from 1 to 3. If severity of abnormal findings was not specified in the bronchoscopy report as mild, moderate, or severe, it was given a score of 1. Each anatomic location could only contribute a maximum of 3 points, for a maximum total score of 9. Upper airway diagnoses included in the score were adenoidal hypertrophy, phayngomalacia, and glossoptosis. Tonsillar hypertrophy was not included in the score as a description of incidence and severity showed high interobserver variation; however, it was reported in our description of findings. Laryngeal findings included in the score were epiglottal thickening, arytenoid thickening, and laryngomalacia. Vocal cord thickening was not scored as it was inconsistently recorded. Lower airway pathology scores included tracheomalacia, bronchomalacia, and tracheal or bronchial deposits. Bronchitis was often incidentally observed but not included in the score as it was not considered a static or progressive finding and could occur due to other reasons, e.g., infection. An example of common findings and of the airway severity score is seen in Figure 1.

Statistical Analysis

Summary statistics were computed for subject level clinical and demographic variables. For the airway findings from the bronchoscopy, the proportion of individuals with a condition in at least one evaluation and the total of positive bronchoscopy for each of abnormal findings were computed. The plot of cumulative incidence of lower airway findings was constructed.

Linear models were fitted via generalized estimation equations (GEE), with an exchangeable working correlation matrix to assess the association between airway severity score and age at bronchoscopy, while accounting for the correlation between repeated measures within patient.

We defined the airway disease progression as the slope of the simple linear regression model using the airway severity score as the response, and age at bronchoscopy as the covariate for each one of the observations. The association between airway disease progression and age at start of ERT was assessed using simple linear regression.

All statistical analyses were performed in Excel Professional Plus 2013, Graph pad prism 9.0.1, and R version 4.0.2 (R Core Team, 2018). Complete case analysis was considered, with *p* < 0.05 determining statistical significance.

## 3. Results

Subjects

Twenty-three male subjects (MPS II is an X-linked recessive disorder primarily affecting males) were included in the IV and intrathecal ERT group. All individuals had the severe or neuronopathic form of Hunter syndrome. Most individuals, 17/23, had been diagnosed prior to the age of 3 years old, with a median (range) age of 2.0 (0–4.5) years at diagnosis. The youngest two subjects were diagnosed early due to positive family history. The median age at time of initiation of IV ERT was 2.6 (0.13–4.7) years and median age at first bronchoscopy was 4.7 (2.3–9.7) years. Number of bronchoscopies per subject averaged 5.6 (2–10). Seven subjects whose data were collected as part of a previous study prior to the advent of ERT were used as a control. These subjects were 2.4–12.6 years old at the time of bronchoscopy.

Cross-sectional Bronchoscopy findings

A total of 130 bronchoscopies were included that occurred between August 2011 and February 2020; no bronchoscopies were excluded. No child experienced complications during bronchoscopy. The airway evaluations were almost all abnormal (129 out of 130 bronchoscopies), and all subjects developed an additional airway abnormality through the course of the observed period. Bronchoscopy findings are summarized in Table 1.

Comparisons to ERT-negative subjects

We previously published data concerning airway findings in individuals with MPS II [6]. A subset of seven individuals in this previous study was ERT naïve, and we applied the airway severity score to the bronchoscopy reports of these seven ERT naïve individuals. Linear regression of severity scores against age tended to show worsening with age for each of the groups (Figure 2).

Longitudinal Airway findings

Generalized estimation equations, which account for repeat evaluations for the subjects showed that increasing airway involvement was associated with older age (*p* = 0.0007) (Figure 3). In many individuals, worsening scores were due to increased occurrence of abnormalities in the lower airway over time. Figure 3 shows the increased slope of airway severity score progression in the lower airways compared to upper or laryngeal findings. Specifically, 16 of the 23 subjects had no lower airway pathology described at the time of their first bronchoscopy but 12 of them subsequently developed at least one abnormal lower airway finding. The cumulative occurrence of lower airway findings is shown in Figure 4. To evaluate if age at time of ERT initiation may affect progression of airway disease, we further analyzed for each patient their slope of progression of the airway severity score from first to last bronchoscopy. Individual slopes are shown in Figure 5. Next, a linear model was constructed for slopes of progression against age at time of ERT initiation. There was no correlation between disease progression and age of initiation of ERT, where one year increase in age at the start of ERT led to an expected increase of −0.03 (95% CI: −0.18, 0.12; *p* = 0.7) units in the airway disease progression (Figure 6).

## 4. Discussion

All individuals in our study were found to have upper airway abnormalities. Despite all individuals having a history of adenoidectomy prior to the first bronchoscopy, all were found to have adenoidal hypertrophy on at least one bronchoscopy, with 122 of the total 130 bronchoscopies mentioning adenoidal hypertrophy. While adenoidal hypertrophy is a well-known finding in MPS II [12], the mechanism of tissue reaccumulating is not entirely clear. Pharyngomalacia is also common in our study, with all individuals having pharyngomalacia on at least one bronchoscopy. These findings of persistent upper airway abnormalities correlate with the well-known problem associated with MPS II of upper airway obstruction and obstructive sleep apnea.

Laryngeal pathology was common as well (Table 1), with 65% having a laryngeal abnormality on at least one bronchoscopy. A prior study focusing on mucosal alterations of the larynx and hypopharynx had evaluated a detailed score to standardize findings across different types of MPS [9]. Of 55 individuals studied, 15 had MPS II. The MPS II individuals had an age range of 5–45 years, and a mean of 20.8 years. Their score showed a varied response to ERT. In this study, one MPS II subject was noted who was examined before and after ERT initiation and who did not have any improvement. Our study differed in the broadened assessment of the entire airway as opposed to a detailed focus on the laryngeal anatomy.

We found that lower airway pathology became more common with increasing age. Notably, all the subjects who had bronchoscopy at age 14 years or older showed lower airway involvement (Figure 4B). This suggests that close monitoring for lower airway pathology in individuals with MPS II should be recommended, and that lower airway pathology will continue to be a major cause of morbidity and mortality in this patient population.

The most frequent lower airway abnormalities are described as “tracheomalacia.” Classic tracheomalacia shows dynamic collapse, generally due to weakening of the tracheal cartilage. However, the trachea in individuals with MPS II is narrowed and tortuous, with evidence of GAG deposition, and is overall different in appearance from classic tracheomalacia. The term tracheopathy is a more accurate descriptor in individuals with MPS II, and we would recommend future use of this term for diagnostic clarity.

We developed a score to compare individuals’ airway findings cross-sectionally and longitudinally. The score attempted to be inclusive of all anatomic pathologies while not allowing for one location to be overwhelmingly represented. Upper airway pathology was almost universal **(**Table 1), often severe, and laryngeal and lower airway pathology would have been less impactful in the overall score had we not limited each anatomic location. Another challenge was that certain findings are more clearly identified earlier in the course of disease than others. For example, pharyngeal collapse is more obvious than subtle nodularity in the trachea. The airway severity score was designed to assess anatomic variation for statistical analysis, not the clinical condition, so the score presently is not designed for clinical implementation. Additionally, upper airway abnormalities are amenable to surgical interventions (e.g., adenoidectomy or tracheostomy) which is very difficult for lower airway pathology. Thus, identification of lower airway lesions prior to an individual’s emergency presentation in the setting of an illness is important. For example, the tracheopathy associated with MPS II is likely seen on bronchoscopy before affecting airway function.

In our study, there was a progression of upper and lower airway abnormalities seen on flexible bronchoscopy despite treatment with IV ERT (Figure 3). We also noted heterogeneous progression between individuals as is typical for the overall disease in MPS II. The fact that the age of ERT initiation was not associated with changes in slope of airway disease progression further supports the idea that IV ERT does not eliminate progression of airway involvement but may slow progression. Yet, the lack of systematic studies prior to routine use of ERT does not allow for a direct larger scale comparison of airway progression. Furthermore, even earlier initiation of ERT may prove efficacious in preventing airway pathology [13]. Individuals with MPS II are most often diagnosed due to clinical signs of the disease, with the median age of diagnosis in one study being 3.3 years but newborn screening is needed to improve the age of diagnosis and long-term outcome [14]. 

A study performed in 2016 involving five adult individuals with MPS II, and which included CTs of the chest to evaluate lower airway involvement, showed thickening of the tracheobronchial tree, and narrowing of the lower airways. The individuals in this study were all adults without CNS disease, and who had been on ERT from 0–6 years at the time the study was performed. This study showed that the degree of narrowing predicted morbidity from airway involvement [15]. The findings in this study of persistent airway thickening and narrowing on the CT chest in individuals with MPS II is similar to our bronchoscopic findings in pediatric subjects. A recent study in adult people with MPS included multiple measures to compile a score of airway severity for clinical use in different types of MPS [16]. This score included exam, upper airway endoscopy, imaging, and if possible, lung function testing. The severity of lower airway involvement with tracheopathy could be well demonstrated on the 3D reconstruction of CT scans. The effect of ERT was not assessed in this retrospective study.

The fact that there were ongoing, progressive airway abnormalities in our study despite consistent use of ERT likely indicates additional mechanisms of injury apart from cellular GAG deposition. This includes uncontrolled inflammation and tissue remodeling due to alterations in signal transduction [17,18,19], aberrations in immune function [20], activation of inflammatory cascades [21], and abnormal oxidative stress [22,23]. Further studies into the mechanisms of progressive airway deterioration despite ERT are warranted.

MPS II also predisposes to increased mucus sections in upper and lower airways, a finding that has improved with ERT. The cause of increased secretions is likely a combination of factors including aspiration, infection, and increased inflammation due to lysosomal dysfunction [18]. It has been observed by bronchoscopists at our institution that the volume of secretions and of signs of bronchitis has decreased since the initiation of ERT. A previous study reviewing the clinical improvements associated with ERT showed that 22 of the 22 subjects studied had improvement in frequency of respiratory infections [24]. Our expectation is that this decrease in mucus burden represents substantial clinical benefit for people with MPS II, though this was not directly measured in our study given the different possible contributors to increased secretions.

The subjects in our study have been receiving intrathecal ERT throughout. This is a potentially confounding factor. However, based on our understanding of the somatic manifestations of MPS II, and the action of idursulfase, we would not expect the addition of intrathecal ERT to substantially affect the airway pathology. However, it is possible that increased airway tone due to improvement in neurologic function would cause improvements in airway patency.

Our study was limited by variability among reporters, as data were collected from bronchoscopy reports from multiple providers. We addressed this variability by careful review of the bronchoscopy reports and by developing a bronchoscopy scoring system that addressed different terminology. While our institution has performed a number of bronchoscopies in individuals with MPS II prior to initiation of ERT, the number and bronchoscopies was small, and lacked longitudinal data for each ERT-naïve subject [6]. A larger subset of ERT-naïve individuals with similar longitudinal observations would have improved comparisons; however, as ERT is now the standard of care, it would be difficult to find a cohort of individuals who had undergone bronchoscopy prior to ERT initiation. Individuals with MPS II can develop antibodies to idursulfase. These antibodies are associated with increased GAG in treated patients and are associated with attenuation of the improvement seen in pulmonary function [25,26]. The efficacy of ERT in preventing airway pathology may be affected by these antibodies. However, we were not able to evaluate anti-idursulfase antibodies’ effects on airway pathology in this study since the antibody data are not available to the authors.

To enhance comparability in reporting MPS II-specific airway anomalies among bronchoscopists, we suggest that providers review the potential airway findings of MPS prior to bronchoscopies. This would enhance identification of subtle manifestations of MPS II such as vocal cord thickening or mucosal nodularity. It is possible that bronchoscopists without experience in MPS II populations may not recognize certain findings as developing pathologies. 

The strengths of this study were its unprecedented access to a large number of bronchoscopies which examined the upper and lower airway, and which were consistently performed yearly over extended periods of time. The original study from which these data were collected included visits for intrathecal injection of enzyme (idursulfase-IT), as such compliance with enzyme replacement therapy treatment for the individuals included was excellent, and we could confidently rule out adherence to therapy as a potential source for error.

## 5. Conclusions

In individuals with mucopolysaccharidosis II, treatment with intravenous ERT did not prevent progression or development of lower airway pathology. Individuals with MPS who are receiving ERT will continue to have progressive airway disease and will need continued monitoring and intervention by their medical providers. Future studies investigating the role of inflammation and other mechanisms of airway disease and the effect of ERTs in the airway are warranted. We are hopeful that widespread adoption of newborn screening for MPS II with early initiation of ERT will more effectively prevent airway pathology in this population.

## Figures and Tables

**Figure 1 jcm-12-00480-f001:**
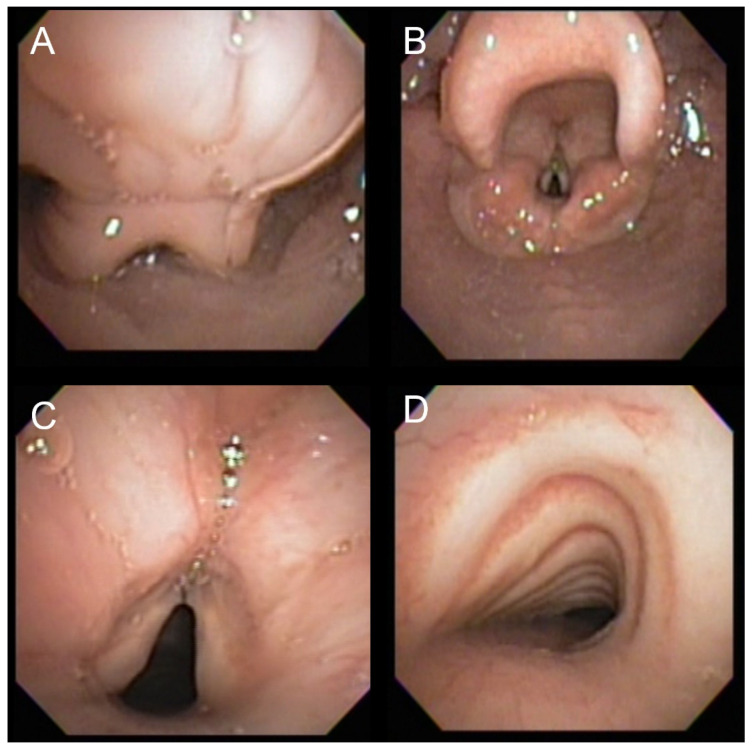
Airway scoring. (**A**) Shows moderate pharyngeal collapse, which would be +2 on the score. (**B**) Shows epiglottal thickening which would be +1 on the score, and mild arytenoid thickening, which would be +1 on the score. (**C**) Shows vocal cord thickening, which is not a finding incorporated into the airway severity score, but is clinically relevant and was inconsistently reported. (**D**) Shows moderate tracheopathy which would be a +2 on the score. Therefore, the total airway severity score for this bronchoscopy would be 6.

**Figure 2 jcm-12-00480-f002:**
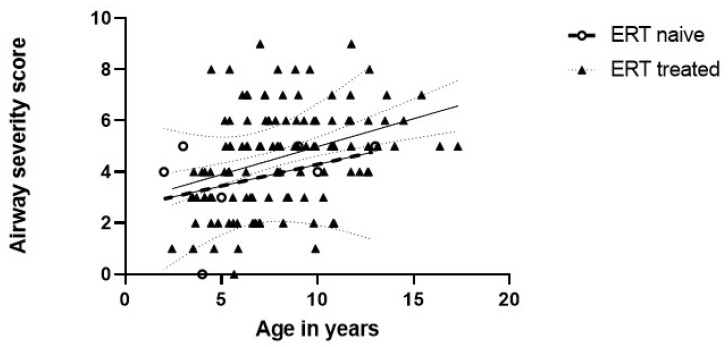
Linear model fitted via generalized estimation equations (GEE) between average airway severity score and age, controlling for ERT group. One year increase in age leads to an expected increase of 0.25 (95% CI: 0.16, 0.33; *p* < 0.001) units in the airway severity score. The dotted lines represent 95% confidence intervals. Open circles are ERT naïve, and closed triangles are ERT treated individuals.

**Figure 3 jcm-12-00480-f003:**
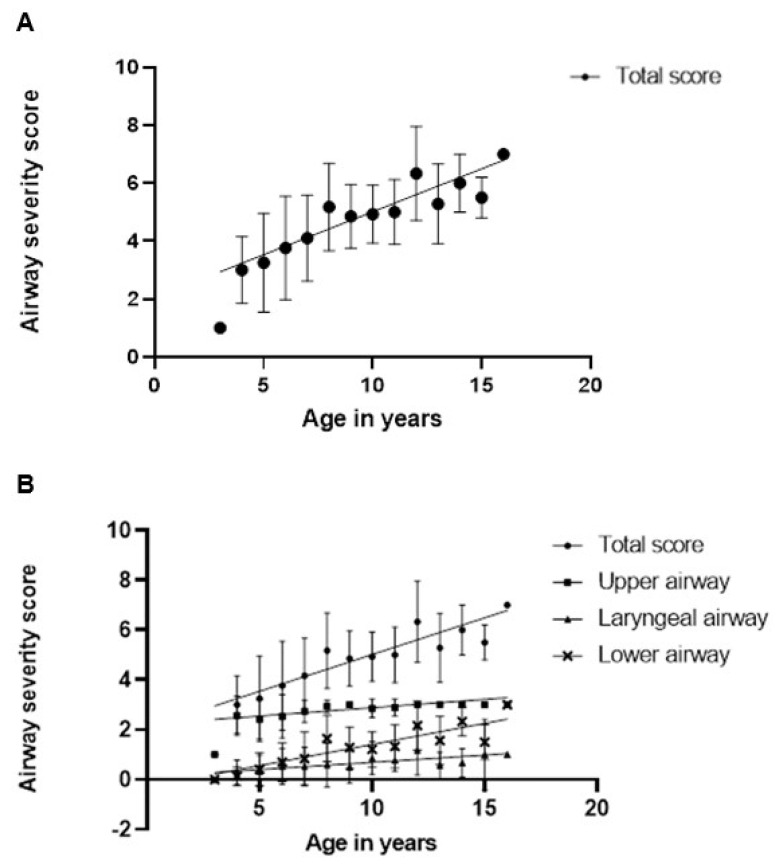
(**A**) Linear model fitted via generalized estimation equations (GEE) between average airway severity score and age. One year increase in age leads to an expected increase of 0.29 (95% CI: 0.2, 0.38; *p* < 0.0001) units in the airway severity score. (**B**) Linear model fitted via GEE between average severity score and age for each anatomic division used in the study. One year increase in age leads to an expected increase of 0.06 for upper airway (95% CI: 0.03, 0.09; *p* < 0.0001) 0.05 for laryngeal airway (95% CI: 0.01, 0.1; *p* < 0.0091) for lower airway 0.16 (0.11, 0.22; *p* < 0.0001) (**A**,**B**) Each point is the combined average airway severity score for bronchoscopies of subjects within this age group in increments of 1 year. Error bars represent standard deviation from the mean.

**Figure 4 jcm-12-00480-f004:**
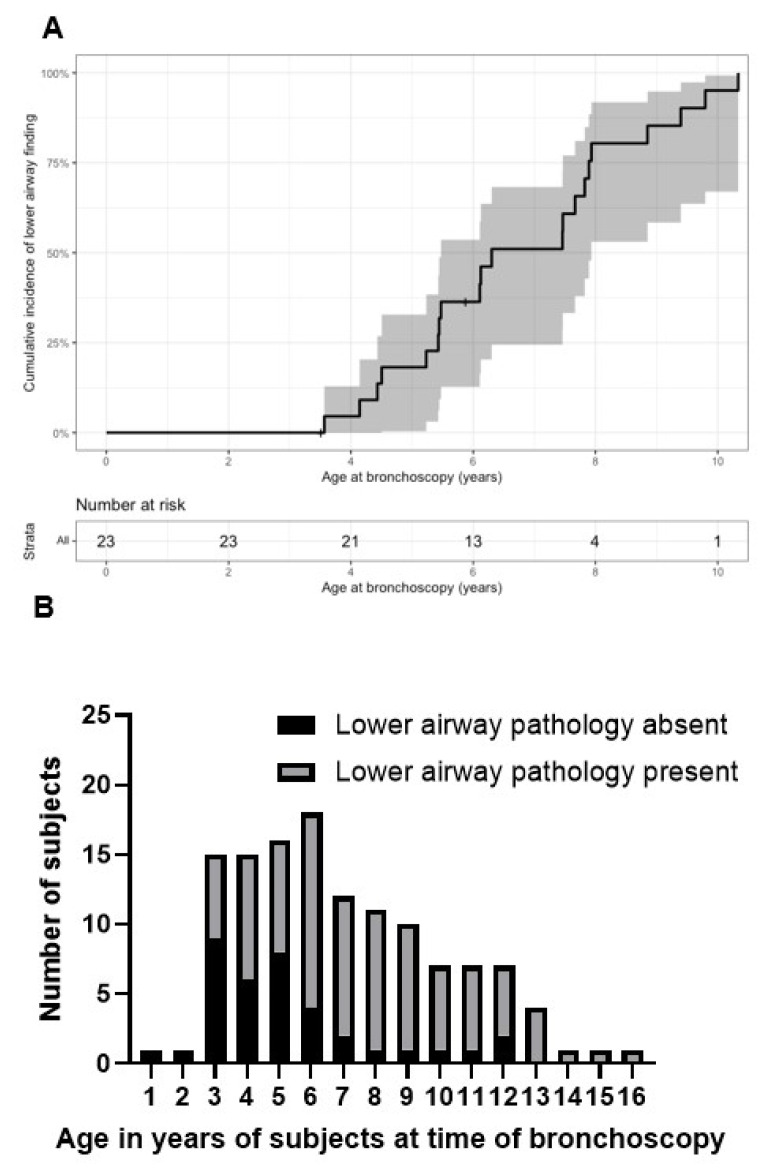
(**A**) Cumulative incidence of lower airway finding with 95% confidence interval. Each deflection point is the entire number of subjects who have bronchoscopies at that age, and the percentage of those subjects which have had bronchoscopies showing lower airway pathology. (**B**) Presence of lower airway findings over time. The dark bars represent bronchoscopies where lower airway pathology is absent, the light bars represent bronchoscopies where lower airway pathology is present. The total number of subjects on the y axis is the number of subjects who underwent bronchoscopy per year of age. Each individual was accounted for only once per year.

**Figure 5 jcm-12-00480-f005:**
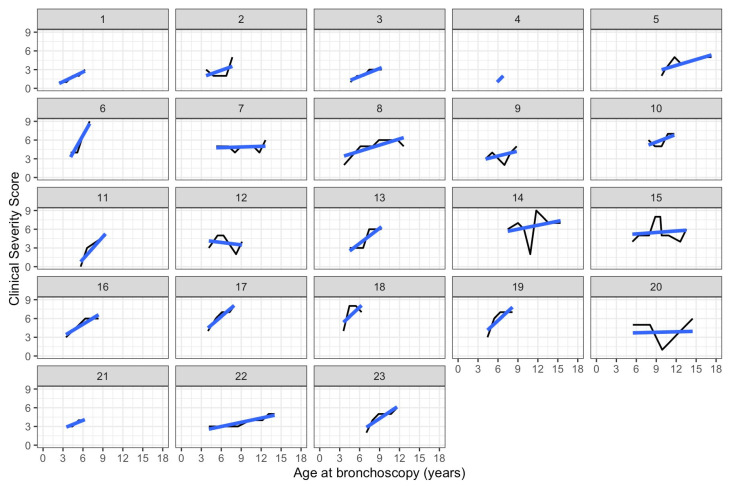
Individual slope of progression of airway disease, expressed as the airway severity score for each subject.

**Figure 6 jcm-12-00480-f006:**
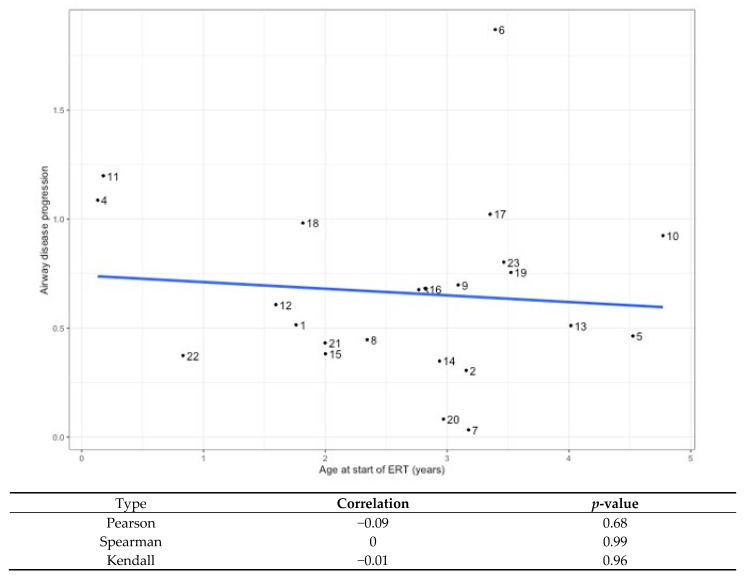
A linear model was fitted with changes in airway severity score as response and age at bronchoscopy as covariate for each one of the *n* = 23 observations. The number indicates subject ID. The association between airway disease progression and age at start of ERT did not reach statistical significance, as one year increase in age at start of ERT leads to an expected increase of −0.03 (95% CI: −0.18, 0.12; *p* = 0.7) units in the airway disease progression.

**Table 1 jcm-12-00480-t001:** Abnormal airway findings diagnosed via bronchoscopy.

Abnormal Findings	Number of Total Bronchoscopies Positive for Condition (% Positive)	% of Individuals Positive for Condition on at Least One Bronchoscopy
**Upper airway**	129/130 (99)	100
Adenoidal hypertrophy	122/130 (93)	100
Pharyngomalacia	114/130 (87)	100
Glossoptosis	40/130 (30)	52
**Laryngeal**	60/130 (46)	65
Laryngomalacia	29/130 (22)	52
Arytenoid thickening	6/130 (5)	22
Epiglottis thickened	35/130 (27)	26
Vocal cord thickening	6/130 (5)	17
**Lower airway**	84/130 (64)	95
Tracheomalacia/tracheopathy	72/130 (55)	86
Bronchomalacia/nodularity	24/130 (18)	54

Legend Table 1: Data are presented for each bronchoscopy including repeat bronchoscopies in subjects with multiple procedures. The column on the left described the proportion of every bronchoscopy performed which was positive for each listed condition. The column on the right describes the percentage of subjects who had the listed finding on at least one evaluation.

## Data Availability

Data sharing is not available due to the possibility of identifying research subjects.

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
