# Peer review of "Airway Findings in Patients with Hunter Syndrome Treated with Intravenous Idursulfase"

_jcm, 2023, doi:10.3390/jcm12020480_

Round 1

Reviewer 1 Report

The authors developed an interesting paper about the fibrobronchoscopic findings in patients with Hunter's disease treated with idursulfase. 

1) Since the study is a retrospective analysis of individuals with the severe or neuronopathic form of MPS II, who participated in clinical trials of intrathecal enzyme replacement, I think it should be clearly stated that all of them were additionally treated with intravenous idursulfase.

2) The authors refer that a limitation of the study is that there were several endoscopists and that in order to improve the comparability of specific MPS II airway abnormalities among bronchoscopists, the authors suggested to them that they review possible MPS airway findings prior to bronchoscopies; might the authors inform if bronchoscopists were blinded to the findings and scores of previously performed endoscopies? 

Author Response

1) Since the study is a retrospective analysis of individuals with the severe or neuronopathic form of MPS II, who participated in clinical trials of intrathecal enzyme replacement, I think it should be clearly stated that all of them were additionally treated with intravenous idursulfase.

Response 1: Thank you for your thoughtful review. The manuscript has been changed to make the administration of IV and intrathecal ERT clearer. 

2) The authors refer that a limitation of the study is that there were several endoscopists and that in order to improve the comparability of specific MPS II airway abnormalities among bronchoscopists, the authors suggested to them that they review possible MPS airway findings prior to bronchoscopies; might the authors inform if bronchoscopists were blinded to the findings and scores of previously performed endoscopies? 

Response 2: The bronchoscopy reports used for this study were obtained retrospectively. The bronchoscopists had full access to the patients' charts including previous bronchoscopy notes. They did not have access to the airway severity score as this was only applied retrospectively for this study. In the discussion we recommended bronchoscopists review some of the unique airway pathology associated with MPS II as to not miss some of the more subtle findings. As our providers have quite a bit of experience with this condition and still reported some of the more subtle findings inconsistently. This is also one of the reasons we added a figure with images representing these findings, as a resource for bronchoscopists. We will make this point in the discussion more clearly. 

Reviewer 2 Report

The authors report findings from a study aimed at improving understanding of the changes over time airway disease in individuals who have Hunter syndrome and the impact of intravenous enzyme replacement therapy (ERT) on changes and progression of airway disease in this patient population.  Airway disease in individuals with Hunter syndrome includes macroglossia, tracheopathy, pharyngomalacia, and adenoid hypertrophy. This study evaluated retrospective data from bronchoscopies in 23 patients with Hunter syndrome who were receiving intravenous ERT (standard of care for Hunter syndrome treatment) while they were enrolled in a clinical trial of intrathecal ERT. The authors report finding from a total of 130 bronchoscopies (obtained prior to general anesthesia for specific clinical trial evaluations) performed in the 23 subjects.  Subjects in the study were between the ages of 2 years and 17 years and had been on IV ERT for at least 6 months (phase I/II trial of IT ERT) or at least 4 months ([phase III/IV trial of IT ERT).  Age of initiation of ERT ranged from 0.13-4.7 years (median 2.6 years). 

Comparative group: The findings were compared to 7 treatment naïve patients from a previous study.

Upper airway disease findings were notable for adenoid hypertrophy in 93% of the bronchoscopies and pharyngomalacia in 87% of the bronchoscopies.

Although most experts in the lysosomal disease community currently are proponents of earlier initiation of ERT treatment in order to optimize clinical outcomes, this study found no association between earlier age of initiation of IV ERT and progression of the airway disease.     

Airway severity score increased with increasing age and presence of lower airway disease increased with increasing age.

Figure 2 shows comparison of airway severity scores in 7 treatment naïve patients (age) at different ages compared to the 23 subject receiving IV ERT (and IT ERT). It appears the youngest treatment naïve child was evaluated at age 2.5 years and the oldest treatment naïve child at approximately age 13 years. One treatment naïve child is distinguished in showing an airway severity score of zero at approximately age 4 years. 

This study’s findings are important in highlighting an unmet need of IV ERT. ERT has been associated with reduced respiratory infections (reference: Lampe C, Bosserhoff AK, Burton BK, Giugliani R, de Souza CF, Bittar C, Muschol N, Olson R, Mendelsohn NJ. Long-term experience with enzyme replacement therapy (ERT) in MPS II patients with a severe phenotype: an international case series. J Inherit Metab Dis. 2014 Sep;37(5):823-9. doi: 10.1007/s10545-014-9686-7. Epub 2014 Mar 5. PMID: 24596019; PMCID: PMC4158409.Despite previous findings of decreased respiratory infections in response to IV ERT,  this study brings to light a notable progression of lower airway disease with age, despite IV ERT. This is concerning, especially in light of findings in previous studies that respiratory failure is the leading cause of death in patients with Hunter syndrome. (references: 1) Lin HY, Chuang CK, Huang YH, Tu RY, Lin FJ, Lin SJ, Chiu PC, Niu DM, Tsai FJ, Hwu WL, Chien YH, Lin JL, Chou YY, Tsai WH, Chang TM, Lin SP. Causes of death and clinical characteristics of 34 patients with Mucopolysaccharidosis II in Taiwan from 1995-2012. Orphanet J Rare Dis. 2016 Jun 27;11(1):85. doi: 10.1186/s13023-016-0471-6. PMID: 27349225; PMCID: PMC4924312.; 2) Burton BK, Jego V, Mikl J, Jones SA. Survival in idursulfase-treated and untreated patients with mucopolysaccharidosis type II: data from the Hunter Outcome Survey (HOS). J Inherit Metab Dis. 2017 Nov;40(6):867-874. doi: 10.1007/s10545-017-0075-x. Epub 2017 Sep 8. PMID: 28887757.)

Questions for the authors:

1)    Figure 2 shows airway severity scores in 7 treatment naïve patients (ages appear to be between 2.5 and 13 years) at different ages, and compares these scores to scores of the 23 subject receiving IV ERT (and IT ERT). It appears the youngest treatment naïve child was evaluated at age 2.5 years and the oldest treatment naïve child at approximately age 13 years. One treatment naïve child is distinguished in showing an airway severity score of zero at approximately age 4 years. Do the authors have comments as to whether if this “outlying” score, in a very small control group, might be skewing the overall comparison between treatment naïve subjects and treated subjects?

2)    Neutralizing anti-ERT antibodies have shown ability to significantly interfere with efficacy of IV ERT.  Were anti-ERT antibodies, and specifically neutralizing antibodies, taken into consideration?  If so, were any associations between antibodies and/or neutralizing antibodies and airway severity score in the IV ERT recipients?

Author Response

1)    Figure 2 shows airway severity scores in 7 treatment naïve patients (ages appear to be between 2.5 and 13 years) at different ages, and compares these scores to scores of the 23 subject receiving IV ERT (and IT ERT). It appears the youngest treatment naïve child was evaluated at age 2.5 years and the oldest treatment naïve child at approximately age 13 years. One treatment naïve child is distinguished in showing an airway severity score of zero at approximately age 4 years. Do the authors have comments as to whether if this “outlying” score, in a very small control group, might be skewing the overall comparison between treatment naïve subjects and treated subjects?

Response 1: Thank you for your very thoughtful review. The control group is unfortunately small, this score of 0 is absolutely an outlier which likely skews the control group lower. However, when this subject is removed from the dataset and the data is analyzed again there is not a significant difference in the outcome of the analysis. 

2)    Neutralizing anti-ERT antibodies have shown ability to significantly interfere with efficacy of IV ERT.  Were anti-ERT antibodies, and specifically neutralizing antibodies, taken into consideration?  If so, were any associations between antibodies and/or neutralizing antibodies and airway severity score in the IV ERT recipients?

Response 2: This point regarding neutralizing antibodies is well made. We did discuss the importance of Anti-ERT antibodies but were not able to perform a suitable analysis of this question due to unavailability of the required data. We agree this would be an interesting question to evaluate and will add a comment on this subject to the manuscript.